# Improvements in Obstetric and Newborn Health Information Documentation following the Implementation of the Safer Births Bundle of Care at 30 Facilities in Tanzania

**DOI:** 10.3390/healthcare12030319

**Published:** 2024-01-26

**Authors:** Benjamin Anathory Kamala, Hege Ersdal, Robert Moshiro, Estomih Mduma, Ulrika Baker, Godfrey Guga, Jan Terje Kvaløy, Dunstan R. Bishanga, Felix Bundala, Boniphace Marwa, Thomas Rutachunzibwa, Japhet Simeo, Honoratha Faustine Rutatinisibwa, Yudas Ndungile, Damas Kayera, Florence Salvatory Kalabamu, Paschal Mdoe

**Affiliations:** 1Department of Research, Haydom Lutheran Hospital, Haydom P.O. Box 9000, Tanzania; moshiror@gmail.com (R.M.); estomduma@gmail.com (E.M.); godfrey.guga@haydom.co.tz (G.G.); pfmdoe@gmail.com (P.M.); 2School of Public Health and Social Sciences, Muhimbili University of Health and Allied Sciences, Dar es Salaam P.O. Box 65001, Tanzania; dbishanga@gmail.com (D.R.B.); bundala.felixambrose@yahoo.com (F.B.); mkalabamu@gmail.com (F.S.K.); 3Faculty of Health Sciences, University of Stavanger, 4021 Stavanger, Norway; hege.ersdal@safer.net; 4Department of Anesthesia, Stavanger University Hospital, 4011 Stavanger, Norway; 5UNICEF Tanzania Country Office, Dar es Salaam P.O. Box 4076, Tanzania; ubaker@unicef.org; 6Department of Mathematics and Physics, University of Stavanger, 4021 Stavanger, Norway; jan.t.kvaloy@uis.no; 7Department of Research, Stavanger University Hospital, 4011 Stavanger, Norway; 8Ifakara Health Institute, Dar es Salaam P.O. Box 78373, Tanzania; 9Ministry of Health, Dodoma P.O. Box 743, Tanzania; 10President’s Office, Regional Administration and Local Government, Dodoma P.O. Box 1923, Tanzania; marwaboniphace3@gmail.com (B.M.); thomruta@yahoo.com (T.R.); japhet.simeo@tanga.go.tz (J.S.); honoratha.rutatinisibwaa@tabora.go.tz (H.F.R.); yudas.ndungile@gmail.com (Y.N.); dajkay@gmail.com (D.K.); 11Department of Pediatrics, Hubert Kairuki Memorial University, Dar es Salaam P.O. Box 65300, Tanzania

**Keywords:** documentation, data missingness, data quality, Safer Births Bundle of Care, quality of care, mentorship, supervision

## Abstract

This paper examines changes in the completeness of documentation in clinical practice before and during the implementation of the Safer Births Bundle of Care (SBBC) project. This observational study enrolled parturient women with a gestation age of at least 28 weeks at the onset of labour. Data collectors extracted information from facility registers and then a central data manager summarised and reported weekly statistics. Variables of clinical significance for CQI were selected, and the proportion of non-documentation was analysed over time. A Pearson chi-square test was used to test for significant differences in non-documentation between the periods. Between 1 March 2021 and 31 July 2022, a total of 138,442 deliveries were recorded. Overall, 75% of all patient cases had at least one missing variable among the selected variables across both periods. A lack of variable documentation occurred more frequently at the district hospital level (81% of patient cases) and health centres (74%) than at regional referral hospitals (56%) (*p* < 0.001). Non-documentation decreased significantly from 79% to 70% after the introduction of the SBBC (*p* < 0.001). A tendency towards negative correlations was noted for most variables. We noted an increased attention to data quality and use which may have a positive impact on the completeness of documentation. However, halfway through the project’s implementation, the completeness of documentation was still low. Our findings support the recommendation to establish short-spaced feedback loops of locally collected data using one data platform.

## 1. Introduction

Globally, neonatal mortality has dropped significantly from 33 to 18 deaths per 1000 live births from 1990 to 2022 [1]. However, in 2022, neonatal mortality in Sub-Saharan Africa was still high at 27 deaths per 1000 live births, 11 times higher than high-income countries [2]. In 2022, 2.3 million deaths occurred in the first month of life, the majority on the day of birth, and more than 2 million stillbirths occurred [2,3]. UN inter-agency estimate reports showed that maternal mortality ratios have also declined from 339 (in 2000) to 223 (in 2022) deaths per 100,000 live births in the same period [4]. However, rates in Sub-Saharan Africa are still high at 536 deaths per 100,000 live births. With current trends, the sustainable development goals (SDG 3.1 and SDG 3.2) aimed at reducing maternal and ending preventable newborn deaths are not likely to be achieved [5].

Global reports show that poor-quality care accounts for 61% of neonatal deaths and half of maternal deaths [6]. To improve the survival rate of mothers and their babies, data on cause-specific morbidity and mortality are needed to guide interventions and policies aimed at improving care [7]. High-quality maternal and perinatal registries which are essential sources of data needed to understand pregnancy and birth outcome trends at facility levels [3] are common in many middle- and high-income countries but scant in most low-income countries [8,9]. Such registries are also valuable for epidemiological research on the factors and causes of maternal and perinatal deaths [10]. The ability of policymakers, health managers, and health facilities to plan, monitor, evaluate, and redesign interventions is highly dependent on accurate and consistent data registration and collection and the ability to make the results easily available and accessible for direct users and decision makers [8,11].

A well-functioning health management information system (HMIS) is one of the World Health Organization’s five health system building blocks that will ensure the production, analysis, dissemination, and use of reliable and timely health information on health determinants, health systems’ performance, and health status [12]. Currently, there are incomplete vital registration systems and poor-quality data on causes of death, leading to biased estimates of the burden in low-income countries [13,14,15,16]. For example, a data validation study in Vietnam found that only 25% of neonatal mortality was reported, leading to inadequate investment in maternal and newborn health programmes [14]. Robust, vital registries and trustworthy routine health information systems are prerequisites for public reports.

In Tanzania, facility-based routine medical records [17,18] are abstracted from the facility registers and uploaded into the national health information database, namely, the district health information system (DHIS-2) [19]. However, the system faces data quality challenges including low accuracy, poor reliability, incomplete records, and delayed data registration [13,17].

This study is nested within the implementation of a three-year continuous quality improvement (CQI) project, the “Safer Births Bundle of Care” (SBBC) project [20]. The rollout in 30 healthcare facilities in five regions is led by Haydom Lutheran Hospital in close collaboration with the Ministry of Health and UNICEF in Tanzania [9,10,11,12,13,14,15,16,17,18,19,20,21,22]. During the implementation of the SBBC package, it has become clear that data are not always captured as intended and as needed, compromising their potential to guide improvement. This gap in data capturing has also been reported in other studies [13,17,21]. The objectives of this study are, therefore, to describe any changes in trends in documentation before and during the implementation of the SBBC and to better understand factors associated with incomplete documentation.

## 2. Materials and Methods

### 2.1. Design, Management, Sites, and Population

The SBBC is a stepped-wedge cluster randomised quality improvement study (ISRCTN Registry: ISRCTN30541755). Haydom Lutheran Hospital, in collaboration with UNICEF Tanzania, the Ministry of Health, the President’s Office for Regional and Local Government, professional bodies (Paediatric and Midwifery Associations), SAFER at Stavanger University Hospital, and Laerdal Global Health, is responsible for implementing SBBC in 30 health facilities in five regions of Tanzania, namely, Manyara, Tabora, Geita, Shinyanga, and Mwanza. All the facilities are categorised as Comprehensive Emergency Obstetric and Newborn Care facilities. They represent different levels of the healthcare system, regional referral hospitals, district hospitals, and health centres. The annual number of births ranges from 1000 to 10,000 among sites, totalling over 100,000 annual deliveries from all the project facilities. The study enrols all parturient women with a live foetus of gestational age 28 weeks and above at the start of labour and seeking delivery services from one of the 30 health facilities.

The rollout of the SBBC follows a stepped-wedge cluster randomised implementation design, implemented from March 2021 to December 2023. Randomisation was conducted using simple random sampling, apart from the first-named region, Manyara, which was purposively selected for logistical and strategic reasons, whereas subsequent regions were chosen randomly. Figure 1 presents an overview of the regions, facilities, and periods of data collection for pre-implementation (acting as the baseline) and post-implementation. Due to the early trends of lower mortality in the first implementation regions, increased understanding of implementation requirements among the SBBC team, and requests from the regional health authorities, Mwanza was allowed to start implementation around the same time as Shinyanga, which was two months earlier than originally scheduled.

### 2.2. The SBBC Project and Implementation Strategy

The overall objective of SBBC is to reduce 24 h newborn mortality by 50%, fresh stillbirths by 20%, and maternal mortality by 10% [20]. The SBBC interventions target emergency obstetric and newborn care (EmONC), i.e., labour monitoring, prevention, and management of postpartum haemorrhage, management of difficult delivery, and resuscitation of a non-breathing newborn using systematic in situ LDHF simulation-based training. The training is aimed at skills acquisition and retention. It further emphasises the importance of record keeping and the use of locally collected data for local continuous quality improvement. The training cascade starts with the training of national facilitators (for 12 days) and the training of facility-based champions (for six days) and HCWs (for 5 days). These initial training sessions are conducted once at the start of the project implementation in each region. The facility champions use the locally captured data and reports/dashboard to conduct periodic debriefing meetings with HCWs, helping them to reflect on and continuously improve their quality of clinical care. Furthermore, they facilitate frequent on-site simulation training focusing on areas in need. Reinforcing HCWs’ competencies through frequent simulation-based training and providing specific and objective data, highlighting areas needing improvement, are likely to motivate and guide HCWs on how to improve the care they provide. This feedback aims to help translate knowledge and skills into clinical practice and establish a culture of excellence within the facilities.

To ensure the facility champions are well supported in their new role, national facilitators, in collaboration with regional and district health management teams, conduct scheduled, supportive supervision and mentorship to provide in-house training and support every quarter. These scheduled mentorship and supportive supervision visits allow for two-way communication to improve practice through skills and experience sharing between mentors/supervisors and mentees/supervisees [17,22]. These visits are engaging, non-intimidating, and non-blaming; they focus on improvement with appropriate follow-up on the gaps identified in the previous visits and give timely feedback on the quality of their previous documentation and reports. Moreover, the SBBC teams (conducting supervision and mentorship) have received SimBegin simulation-based training: https://www.safer.net/simbegin/ (accessed on 10 March 2023) which focuses on constructive facilitation, teamwork, and debriefing models. At each facility, HCWs are given time to reflect on their activities, they learn to appreciate what is performed well, and then discuss things they would like to change and/or do better in subsequent periods. Together they create “takeaway messages” from patient case reports that have been discussed.

### 2.3. Data Collection and Management

In Tanzania, HMIS data are predominantly related to service delivery and are collected at all levels of the health system. Data from individual health facilities are sent in aggregate form to the district level, generating a summary of indicators. Data are then transmitted from the district to the regional and national levels.

Early in the implementation of the SBBC, in January 2021, 60 data collectors (2 for each site) and five regional coordinators (one for each region) were trained to conduct and oversee data collection. Data are recorded primarily by facility HCWs, who are full-time members of the facility staff. The data collectors are independent (i.e., not members of hospital staff) and collect the registered data from patient case notes (primarily partograph), labour and delivery registers, and perinatal audit reports, usually documented by facility staff in their routine patient care. If any information is missing from the above source documents, the field is left blank. The data collectors also manage quality control procedures, including the correction of any errors before transfer to the central server at Haydom.

Prospective baseline data collection started at all sites on 1 March 2021. The data collectors entered the relevant indicators of labour and newborn care and maternal and newborn characteristics and health outcomes into an electronic data collection system (i.e., Open Data Kit (ODK)) daily. Data were uploaded to the central secure servers daily and checked by the data manager at Haydom, who performed a final quality check before storage. Any errors/queries identified were sent back to the regional coordinators and responsible data collectors for resolution. The central data manager summarised basic statistics reported to each facility weekly and shared this with the team of investigators and HCWs for reviews and CQI at each of the health facilities.

### 2.4. Variables Included in This Analysis

The SBBC case report form contains about 60 variables. In this study, all variables were included for analysis and frequency tables were generated. The variables that were most frequently missing were identified first, then the most clinically relevant indicators were selected by a multidisciplinary team. Eight childbirth conditions and practice-related variables were included for further analyses, based on clinical importance and a high level of non-documentation. These variables were gestation age (in weeks), amniotic fluid colour (clear, meconium stained, or blood-stained), duration of first and second stages of labour (in minutes), foetal heart rate monitoring during labour documented (yes/no), antenatal care (ANC) attendance (yes/no), ANC problems (yes/no), and presentation of the baby at birth (cephalic, breech). Moreover, the three essential patient outcome variables were included: birth outcome at 30 min (i.e., alive, fresh stillbirth, or macerated stillbirth), neonatal outcome at 24 h (i.e., normal, seizures, dead, or admitted to neonatal unit), and the maternal outcome at discharge (i.e., normal, maternal near miss, or maternal death). Lack of documentation was defined as a particular variable that was not recorded in either of the source documents and thus appeared blank in the Open Data Kit (ODK) database regardless of whether the clinical event occurred or not.

Additionally, the number of HCWs in each maternity ward was divided by the average number of births per month in the same facility to arrive at the workload figure per HCW per site. This estimate enabled an analysis of the relationship between the proportions of missing data documentation and the workload per HCW per site, both before and during implementation.

### 2.5. Statistical Methods

Counts and proportions were used to show levels of missing documentation, displayed in tables and graphs to show the differences between the time points (baseline vs. after the start of SBBC implementation), regions, and facility levels. Timeline charts were used to show trends of missing documentation in the months from March 2021 to July 2022 at both regional and facility levels. The Pearson chi-square test at the 0.05 level was used to test for significant differences between the two time periods (before vs. after the start of SBBC implementation). Pearson correlation was used to assess the relationship between the proportion of HCWs and the proportion of data missingness across the facilities. The data were analysed using R version 4.2.1 software.

### 2.6. Ethical Considerations

The SBBC is approved by the National Institute for Medical Research in Tanzania (Ref. NIMR/HQ/R.8a/Vol.IX/3458) and the Regional Ethical Committee in Norway (Ref. 229725). Permission to publish was obtained from NIMR Ref. No: NIMR/HQ/P.12 VOL XXXV/90. All women admitted to the labour ward for delivery were informed about the quality improvement project and the new clinical tools. Before starting the project, all research assistants and investigators were trained in good clinical practice, research ethics, research integrity, and confidentiality. All data were managed and stored according to the governing laws in Tanzania. All data were de-identified so that individual confidentiality was maintained.

## 3. Results

### 3.1. Overall Findings

In total, 138,442 deliveries were recorded between 1 March 2021 and 31 July 2022, out of which 67,775 deliveries took place in the pre-implementation period and 70,667 deliveries took place after the start of the implementation depending on the timeline allocation as described in Figure 1. Social and demographic characteristics were comparable across the regions. Overall, the level of non-documentation was high across the 11 selected variables, whereby an average of 75% of all patient cases missed documentation of at least one variable across both periods (baseline and after the start of implementation), as seen in Figure 2A.

Documentation improved during the implementation of the SBBC across all regions and all levels of health facilities (Figure 2). Of all case notes in the baseline, 79% missed documentation of at least one of the selected variables, whereas 70% of cases had missing documentation following the baseline period and during SBBC implementation (Table 1 and Figure 2A).

The variables with the most prevalent missing documentation before and after the start of implementation were the duration of the first stage of labour (49%), the gestational age (38%), and amniotic fluid colour (35%) (Figure 2). Overall non-documentation of the key outcome variables, i.e., birth outcomes at 30 min and at 24 h and maternal outcomes, were 0.4%, 2.1%, and 2.4%, respectively, at baseline and decreased to almost zero, i.e., 0%, 0.3%, and 0.2%, respectively, during the implementation period (Figure 3).

### 3.2. Comparison of Non-Documentation across Health Facility Levels

Overall, missing documentation of one or more variables was most prevalent at the district hospital level (81% of patient case notes), followed by health centres (74%) and then regional referral hospitals (55%) (Figure 2B). Documentation of the patient case notes improved significantly across all health facility levels after the start of the SBBC implementation (Table 1). The proportion of missing documentation (one or more variables) declined from 87% to 76% (of patient case notes) at district hospitals, from 78% to 70% at health centres, and from 60% to 48% at regional referral hospitals. Gestational age documentation improved significantly among district hospitals (a decline in missing documentation from 55% to 39%), but there was no significant decline among health centres (from 33% to 32% non-documentation) and regional referral hospitals (from 20% to 19% non-documentation) (Table 1). During the baseline period, newborn outcome data at 30 min was more frequently not documented at the regional referral hospital level (1.9%) compared to district hospitals and health centres, both at 0.1%. Non-documentation of the 30 min outcome declined to near zero during SBBC implementation across all health facility sites/levels. A similar trend of improved documentation was observed for neonatal outcomes at 24 h and maternal outcomes at discharge across all levels of health facilities.

### 3.3. Comparison of Non-Documentation across Regions

The SBBC implementation period was longest in Manyara (13 months), followed by Tabora (11 months) and Geita (9 months), and it was the shortest in Shinyanga and Mwanza (about 6 months each) (Figure 1). At the regional level, during the baseline period, the proportion of missing documentation was highest in Manyara (87%) and Tabora (85%) and lowest in Shinyanga (71%). The decline in missing documentation was highest in Manyara (a 15% improvement) and Tabora (12%) and lowest in Shinyanga (3%) (Figure 3). There was improved documentation of the gestational age variable in all regions, with the highest improvement in Manyara (29%) and the lowest in Shinyanga (0.2%). During the baseline period, foetal heart rate recording was most frequently missing in Tabora (21%) and least problematic in Shinyanga (4%). During the baseline period, the newborn outcomes at 30 min variable was documented in almost all patient cases in Manyara, Tabora, Geita, and Mwanza and missing in 1.4% of cases in Shinyanga (Table 2). During implementation, the newborn outcomes at 30 min variable was recorded in all patient cases across all regions. At baseline, the newborn outcome at 24 h variable was more frequently not recorded in Manyara, i.e., 13% of patient case notes. The variable was less frequently missing in Tabora (2.7%) and least undocumented in Geita (0.2%). During implementation, the level of non-documentation declined to almost zero across all regions (Table 2). Maternal outcome at discharge was most frequently missing in Manyara (9.2% of patient cases), followed by Shinyanga (6.4%) during baseline. Documentation improved significantly after SBBC implementation, to 0.4% and 0.5%, respectively (Table 2).

### 3.4. The Relative Number of HCWs and Case Note Documentation

Figure 4 illustrates the correlations between the proportion of HCWs and the proportion of missing variable documentation across facilities. The first row and first column display the correlation between the relative proportion of HCWs per birth versus the proportion of missing documentation for each of the 11 selected variables, across facilities. A tendency towards negative correlations is noted for most variables, i.e., the more HCWs, the less missing variable documentation. The negative correlation (i.e., more missing documentation with fewer HCWs) is more evident during implementation than during the baseline period (i.e., 10 versus 7 variables out of 11). However, it was only statistically significant for the documentation of foetal presentation and fluid colour during baseline and only for foetal presentation after baseline. The other rows and columns in the figure illustrate correlations in non-documentation between the variables.

## 4. Discussion

In this study, we describe the frequency of non-documentation of 11 key obstetric and newborn indicators and changes in documentation over time in 30 facilities across five regions before and during the implementation of the SBBC CQI. Overall, there was a high proportion of missing data across all sites for all key indicators before the implementation of the SBBC. District hospitals and health centres had more missing data than the regional referral hospitals. However, the proportion of missing data decreased significantly following the introduction of the SBBC. This was true across all the 11 variables, facilities, and regions. However, the proportions of missing data remain unreasonably high despite this improvement, indicating that there are other underlying factors, which would include lack of accountability structures to follow up what is recorded as per the national health information system. Moreover, there was a trend towards a negative correlation between the proportion of missing data and the availability of HCWs, i.e., there were fewer missing variables when there were higher numbers of HCWs.

The non-documentation of variables in public facilities in Tanzania have been reported before [14,15]. However, the extent of non-documentation described in this study is higher than in previous reports where the level of non-documentation in Tanzania was found to be 50% [13]. Factors which may contribute to the high rates of non-documentation include limited understanding and the inadequate analysis and use of data among health workers. Inadequate feedback systems on missing documentation and its consequences may also contribute to non-documentation [23]. A heavy workload, insufficient mentorship, and a lack of supportive supervision are other factors described previously in Tanzania and elsewhere [17,21,24]. Indeed, in all the facilities included in this study, evidence for the routine use of health facility data to guide decisions and facilitate quality improvement was scant at baseline. This may have had a negative influence on HCWs’ perceptions of the value of proper data recording, as they were not using data directly for continuous quality improvement. Several studies in Tanzania and other poorly resourced countries have reported the underuse of routine data at the health facility level [25,26]. Not using routine data at the facility level dissociates HCWs from the data, thereby diminishing their perception of the importance of appropriate documentation [17,25,27,28].

There could be several reasons for the observed improvement in the documentation of data following the SBBC implementation. First, two facility champions (per facility) and HCWs in the facilities were trained on the SBBC package and the importance of data quality, emphasising that their facility synthesised data would be sent back to them on a weekly basis. Second, after initial training at the facilities, both national facilitators and regional coordinators conducted regular mentorship and supportive supervision, respectively. On each visit, data quality and CQI using local data were discussed and emphasised. These mentorship and supervision visits may have offered opportunities for communication between mentors and HCWs to improve practice by sharing skills and experience [17,22]. Third, SimBegin simulation-based training emphasising a “no blame, no shame” culture may have facilitated improved documentation. The training is engaging and emphasises “no blame”, with a focus on closing identified clinical practice gaps, based on HCWs’ data.

At baseline, we found variations in missing data across the regions, with the highest data completeness in Manyara and the lowest in Shinyanga. However, the level of improvement during the implementation phase increased significantly in all regions, with the biggest improvement observed in Manyara and the smallest in Shinyanga. Variations in regional performance in health information data documentation have also been reported in similar studies [13,15,27,29]. The main reason for more improvement in Manyara may be the longest implementation time, as well as having the lowest baseline level of data completeness. Other reasons for the regional differences may include differences in the frequency and quality of supervision and mentorship visits, inadequate feedback on the missing information, and other contextual and health service factors not explored in this study [17,22].

We noted a negative correlation between the proportion of missing data in a facility and the number of HCWs per delivery in that facility. The negative correlation was more profound during the implementation phase than at the baseline. Other studies in poorly resourced settings have reported similar findings [17,21], whereby HCWs’ workload has been reported to hamper the recording, processing, and reporting of health facility data [24]. Tanzania has about a 50% gap in human resources for health [30]. Documentation in the source registers is usually carried out by the same HCWs who are responsible for the management of patients. These registers also contain many variables to be captured, which is considered time-consuming [31]. Moreover, in addition to the registers, depending on the level of the health facility, the HCW may need to complete multiple electronic medical records and other electronic platforms. Hence, while taking care of a mother–baby dyad, the same HCW needs to perform multiple data entries, thereby increasing the workload.

It is important to note that HCWs training under SBBC may have contributed to the observed improvement in documentation enhancing data-driven feedback as well as the provision of more quality health care as it was documented in the recent SBBC halfway publication. In this publication, it was noted that there has been a steady reduction in early newborn and maternal mortality and a fluctuation in reported fresh stillbirths across times and regions [32].

This study was limited by the fact that data used for analysis were taken only halfway through the study period; hence, the findings reported here may change by the end of the implementation period. Moreover, the study included higher centres providing EmONC services in the five regions, which are most likely to be burdened by cases. Nevertheless, comparing data completeness over time within the same facility population addresses any bias. On the other hand, improved documentation reported following SBBC implementation could be augmented because HCWs were aware of being observed, i.e., the Hawthorne effect. Additional studies would be required to explore barriers and facilitators of documentation, as well as determining strategies to sustain the improved data management practices even beyond the study period.

## 5. Conclusions

At baseline, we observed a high proportion of missing data reflecting poor documentation practices which was associated with a high caseload for HCWs. We noted increased attention to data quality and use which may have a positive impact on the completeness of documentation. Potential reasons for this reduction have been discussed in relation to the components implemented as part of the SBBC package. However, halfway through the project, the completeness of implementing documentation was still low and other factors likely contributed to the persistence of this challenge which will be explored from other data sources, such as readiness assessments and supportive supervision reports, at later stages.

Our findings strengthen the recommendation to establish short-spaced feedback loops of locally collected data using one data platform like the existing DHIS-2 dashboard linked to increased accountability measures. This can identify gaps in reporting in a timelier manner and hence improve and sustain proper documentation.

## Figures and Tables

**Figure 1 healthcare-12-00319-f001:**
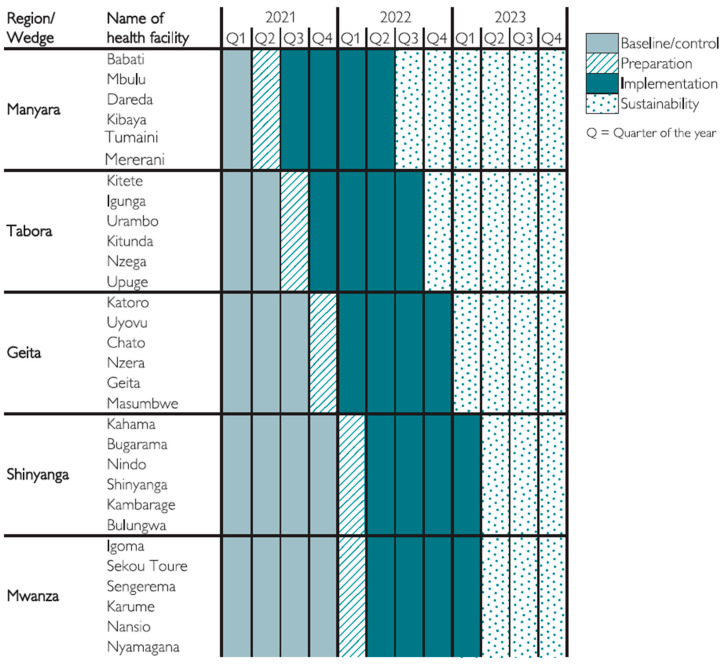
Periods of interventions in the different regions (wedges) and health facilities (clusters) during the rollout of the study.

**Figure 2 healthcare-12-00319-f002:**
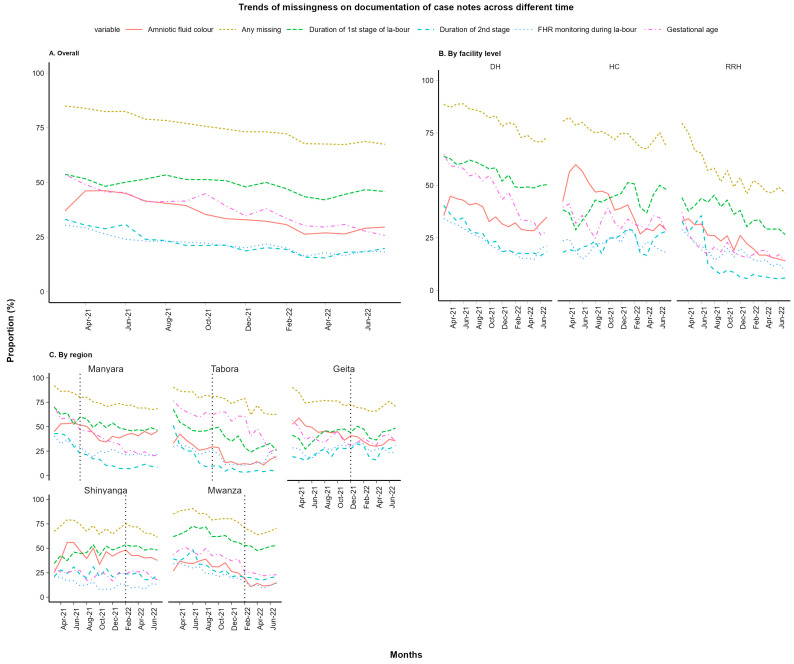
The proportion of missing documentation in patient case notes (**A**) overall, (**B**) by level of health facility, and (**C**) before and after implementation.

**Figure 3 healthcare-12-00319-f003:**
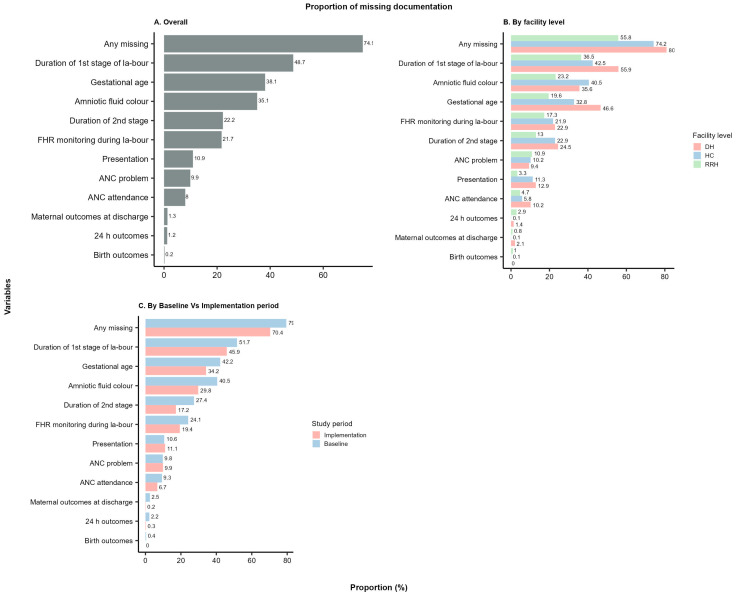
Trends in the proportion of cases with missing documentation of one or more variables before and during implementation of SBBC (**A**) overall, (**B**) by level of health facility, and (**C**) by region.

**Figure 4 healthcare-12-00319-f004:**
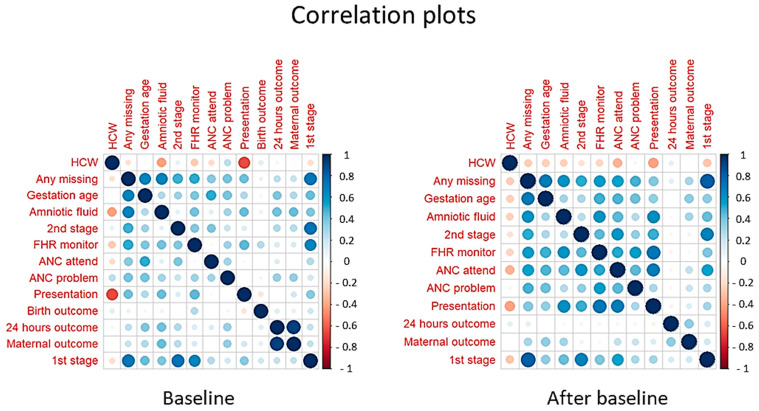
Correlations between the relative proportion of healthcare workers per birth and the proportion of missing documentation for each variable. Note: The colour red indicates a negative correlation and blue is a positive correlation. Darker colours indicate strong correlations (see the colour bar at the right of the plots.) Also, the size of the circle indicates the extent of the correlation; ANC: antenatal care; FHR: foetal heart rate; HCW: health care worker.

**Table 1 healthcare-12-00319-t001:** Comparison of missing documentation between baseline and during implementation by levels of health facilities from March 2021 to July 2022.

All	All Sites		Regional Referral Hospital		District Hospital Level		Health Centre Level	
	Baseline (N = 67,775)	Implementation (N = 70,667)	Baseline (N = 12,059)	Implementation (N = 10,151)	Baseline (N = 34,699)	Implementation (N = 39,699)	Baseline (N = 21,017)	Implementation (N = 20,817)
Any missing *p* < 0.0001								
Missing	53,706 (79.2)	49,463 (70.0)	7278 (60.4)	4857 (47.8)	30,041 (86.6)	30,024 (75.6)	16,387 (78.0)	14,582 (70.0)
Not missing	14,069 (20.8)	21,204 (30.0)	4781 (39.6)	5294 (52.2)	4658 (13.4)	9675 (24.4)	4630 (22.0)	6235 (30.0)
Gestational age *p* < 0.0001								
Missing	28,622 (42.2)	24,168 (34.2)	2419 (20.1)	1935 (19.1)	19,182 (55.3)	15,533 (39.1)	7021 (33.4)	6700 (32.2)
Not missing	39,153 (57.8)	46,499 (65.8)	9640 (79.9)	8216 (80.9)	15,517 (44.7)	24,166 (60.9)	13,996 (66.6)	14,117 (67.8)
Amniotic fluid colour *p* < 0.0001								
Missing	27,517 (40.6)	21,067 (29.8)	3484 (28.9)	1683 (16.6)	13,558 (39.1)	12,947 (32.6)	10,475 (49.8)	6437 (30.9)
Not missing	40,258 (59.4)	49,600 (70.2)	8575 (71.1)	8468 (83.4)	21,141 (60.9)	26,752 (67.4)	10,542 (50.2)	14,380 (69.1)
Duration of 2nd stage of labour *p* < 0.0001								
Missing	18,599 (27.4)	12,160 (17.2)	2353 (19.5)	537 (5.3)	11,770 (33.9)	6520 (16.4)	4476 (21.3)	5103 (24.5)
Not missing	49,176 (72.6)	58,507 (82.8)	9706 (80.5)	9614 (94.7)	22,929 (66.1)	33,179 (83.6)	16,541 (78.7)	15,714 (75.5)
FHR monitoring during labour *p* < 0.0001								
Missing	16,403 (24.2)	13,753 (19.5)	2495 (20.7)	1376 (13.6)	9487 (27.3)	7623 (19.2)	4421 (21.0)	4754 (22.8)
Not missing	51,372 (75.8)	56,914 (80.5)	9564 (79.3)	8775 (86.4)	25,212 (72.7)	32,076 (80.8)	16,596 (79.0)	16,063 (77.2)
ANC attendance *p* < 0.0001								
Missing	6323 (9.3)	4707 (6.7)	726 (6.0)	300 (3.0)	4517 (13.0)	3084 (7.8)	1080 (5.1)	1323 (6.4)
Not missing	61,452 (90.7)	65,960 (93.3)	11,333 (94.0)	9851 (97.0)	30,182 (87.0)	36,615 (92.2)	19,937 (94.9)	19,494 (93.6)
ANC problem *p* = 0.4185								
Missing	6640 (9.8)	7016 (9.9)	1316 (10.9)	1086 (10.7)	3636 (10.5)	3351 (8.4)	1688 (8.0)	2579 (12.4)
Not missing	61,135 (90.2)	63,651 (90.1)	10,743 (89.1)	9065 (89.3)	31,063 (89.5)	36,348 (91.6)	19,329 (92.0)	18,238 (87.6)
Presentation *p* = 0.0183								
Missing	7255 (10.7)	7845 (11.1)	495 (4.1)	243 (2.4)	4823 (13.9)	4828 (12.2)	1937 (9.2)	2774 (13.3)
Not missing	60,520 (89.3)	62,822 (88.9)	11,564 (95.9)	9908 (97.6)	29,876 (86.1)	34,871 (87.8)	19,080 (90.8)	18,043 (86.7)
Birth outcome (at 30 min *p* < 0.0001)								
Missing	281 (0.4)	0 (0.0)	229 (1.9)	0 (0.0)	31 (0.1)	0 (0.0)	21 (0.1)	0 (0.0)
Not missing	67,494 (99.6)	70,667 (100.0)	11,830 (98.1)	10,151 (100.0)	34,668 (99.9)	39,699 (100.0)	20,996 (99.9)	20,817 (100.0)
24 h outcomes *p* < 0.0001								
Missing	1452 (2.1)	193 (0.3)	544 (4.5)	72 (0.7)	876 (2.5)	113 (0.3)	32 (0.2)	8 (0.0)
Not missing	66,323 (97.9)	70,474 (99.7)	11,515 (95.5)	10,079 (99.3)	33,823 (97.5)	39,586 (99.7)	20,985 (99.8)	20,809 (100.0)
Maternal outcome at discharge *p* < 0.0001								
Missing	1653 (2.4)	141 (0.2)	160 (1.3)	16 (0.2)	1450 (4.2)	116 (0.3)	43 (0.2)	9 (0.0)
Not missing	66,122 (97.6)	70,526 (99.8)	11,899 (98.7)	10,135 (99.8)	33,249 (95.8)	39,583 (99.7)	20,974 (99.8)	20,808 (100.0)
Duration of 1st stage of labour *p* < 0.0001								
Missing	35,075 (51.8)	32,435 (45.9)	5093 (42.2)	3020 (29.8)	21,472 (61.9)	20,145 (50.7)	8510 (40.5)	9270 (44.5)
Not missing	32,700 (48.2)	38,232 (54.1)	6966 (57.8)	7131 (70.2)	13,227 (38.1)	19,554 (49.3)	12,507 (59.5)	11,547 (55.5)

Data are shown as number (percent); FHR = Foetal Heart Rate, ANC = Antenatal Care; *p*-values across rows.

**Table 2 healthcare-12-00319-t002:** Comparison of missing documentation between baseline and after baseline by region from March 2021 to July 2022.

All	Manyara		Tabora		Geita		Shinyanga		Mwanza	
	Baseline (N = 4943)	Implementation (N = 15,658)	Baseline (N = 7382)	Implementation (N = 14,365)	Baseline (N = 19,307)	Implementation (N = 19,073)	Baseline (N = 15,545)	Implementation (N = 8453)	Baseline (N = 20,598)	Implementation (N = 13,118)
Any missing *p* < 0.0001										
Missing	4312 (87.2)	11,378 (72.7)	6237 (84.5)	10,334 (71.9)	14,936 (77.4)	13,206 (69.2)	11,042 (71.0)	5733 (67.8)	17,179 (83.4)	8812 (67.2)
Not missing	631 (12.8)	4280 (27.3)	1145 (15.5)	4031 (28.1)	4371 (22.6)	5867 (30.8)	4503 (29.0)	2720 (32.2)	3419 (16.6)	4306 (32.8)
Gestational age *p* < 0.0001										
Missing	3031 (61.3)	5056 (32.3)	4888 (66.2)	7221 (50.3)	7939 (41.1)	6814 (35.7)	3689 (23.7)	1984 (23.5)	9075 (44.1)	3093 (23.6)
Not missing	1912 (38.7)	10,602 (67.7)	2494 (33.8)	7144 (49.7)	11,368 (58.9)	12,259 (64.3)	11,856 (76.3)	6469 (76.5)	11,523 (55.9)	10,025 (76.4)
Amniotic fluid colour *p* < 0.0001										
Missing	2529 (51.2)	6661 (42.5)	2413 (32.7)	2448 (17.0)	9114 (47.2)	6668 (35.0)	6789 (43.7)	3520 (41.6)	6672 (32.4)	1770 (13.5)
Not missing	2414 (48.8)	8997 (57.5)	4969 (67.3)	11,917 (83.0)	10,193 (52.8)	12,405 (65.0)	8756 (56.3)	4933 (58.4)	13,926 (67.6)	11,348 (86.5)
Duration of 2nd stage of labour *p* < 0.0001										
Missing	1928 (39.0)	1923 (12.3)	1861 (25.2)	836 (5.8)	4317 (22.4)	5009 (26.3)	3849 (24.8)	1832 (21.7)	6644 (32.3)	2560 (19.5)
Not missing	3015 (61.0)	13,735 (87.7)	5521 (74.8)	13,529 (94.2)	14,990 (77.6)	14,064 (73.7)	11,696 (75.2)	6621 (78.3)	13,954 (67.7)	10,558 (80.5)
FHR monitoring during labour *p* < 0.0001										
Missing	1733 (35.1)	3615 (23.1)	2012 (27.3)	2283 (15.9)	5019 (26.0)	5275 (27.7)	2155 (13.9)	963 (11.4)	5484 (26.6)	1617 (12.3)
Not missing	3210 (64.9)	12,043 (76.9)	5370 (72.7)	12,082 (84.1)	14,288 (74.0)	13,798 (72.3)	13,390 (86.1)	7490 (88.6)	15,114 (73.4)	11,501 (87.7)
ANC attendance *p* < 0.0001										
Missing	708 (14.3)	970 (6.2)	1442 (19.5)	579 (4.0)	1256 (6.5)	1529 (8.0)	692 (4.5)	893 (10.6)	2225 (10.8)	736 (5.6)
Not missing	4235 (85.7)	14,688 (93.8)	5940 (80.5)	13,786 (96.0)	18,051 (93.5)	17,544 (92.0)	14,853 (95.5)	7560 (89.4)	18,373 (89.2)	12,382 (94.4)
ANC problem *p* = 0.4185										
Missing	936 (18.9)	1472 (9.4)	1273 (17.2)	1348 (9.4)	1461 (7.6)	2681 (14.1)	533 (3.4)	726 (8.6)	2437 (11.8)	789 (6.0)
Not missing	4007 (81.1)	14,186 (90.6)	6109 (82.8)	13,017 (90.6)	17,846 (92.4)	16,392 (85.9)	15,012 (96.6)	7727 (91.4)	18,161 (88.2)	12,329 (94.0)
Presentation *p* = 0.0183										
Missing	259 (5.2)	1157 (7.4)	424 (5.7)	913 (6.4)	2,59 (13.3)	3333 (17.5)	1271 (8.2)	1487 (17.6)	2742 (13.3)	955 (7.3)
Not missing	4684 (94.8)	14,501 (92.6)	6958 (94.3)	13,452 (93.6)	16,748 (86.7)	15,740 (82.5)	14,274 (91.8)	6966 (82.4)	17,856 (86.7)	12,163 (92.7)
Birth outcomes (at 30 min *p* < 0.0001)										
Missing	6 (0.1)	0 (0.0)	11 (0.1)	0 (0.0)	30 (0.2)	0 (0.0)	215 (1.4)	0 (0.0)	19 (0.1)	0 (0.0)
Not missing	4937 (99.9)	15,658 (100.0)	7371 (99.9)	14,365 (100.0)	19,277 (99.8)	19,073 (100.0)	15,330 (98.6)	8453 (100.0)	20,579 (99.9)	13,118 (100.0)
24 h outcomes *p* < 0.0001										
Missing	630 (12.7)	61 (0.4)	212 (2.9)	25 (0.2)	48 (0.2)	4 (0.0)	253 (1.6)	86 (1.0)	309 (1.5)	17 (0.1)
Not missing	4313 (87.3)	15,597 (99.6)	7170 (97.1)	14,340 (99.8)	19,259 (99.8)	19,069 (100.0)	15,292 (98.4)	8367 (99.0)	20,289 (98.5)	13,101 (99.9)
Maternal outcomes at discharge *p* < 0.0001										
Missing	457 (9.2)	61 (0.4)	57 (0.8)	18 (0.1)	53 (0.3)	12 (0.1)	998 (6.4)	41 (0.5)	88 (0.4)	9 (0.1)
Not missing	4486 (90.8)	15,597 (99.6)	7325 (99.2)	14,347 (99.9)	19,254 (99.7)	19,061 (99.9)	14,547 (93.6)	8412 (99.5)	20,510 (99.6)	13,109 (99.9)
Duration of 1st stage of labour *p* < 0.0001										
Missing	3080 (62.3)	7905 (50.5)	3806 (51.6)	5073 (35.3)	7915 (41.0)	8486 (44.5)	7025 (45.2)	4262 (50.4)	13,249 (64.3)	6709 (51.1)
Not missing	1863 (37.7)	7753 (49.5)	3576 (48.4)	9292 (64.7)	11,392 (59.0)	10,587 (55.5)	8520 (54.8)	4191 (49.6)	7349 (35.7)	6409 (48.9)

Data are shown as number (percent); FHR = Foetal Heart Rate, ANC = Antenatal Care; *p*-values across rows.

## Data Availability

Data can be requested from Haydom Lutheran Hospital, P.O. Box 9000 Haydom, Manyara, Tanzania, Tel. +255(0)27 253 3194/5, Fax +255(0)27 253 3734, E-mail post@haydom.co.tz.

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
