# Peer review of "Improvements in Obstetric and Newborn Health Information Documentation following the Implementation of the Safer Births Bundle of Care at 30 Facilities in Tanzania"

_healthcare, 2024, doi:10.3390/healthcare12030319_

Round 1
Reviewer 1 Report
Comments and Suggestions for Authors
The authors present a stepped-wedge cluster randomized trial in five regions of Tanzania with one year implementation period. The objective of the trial is to increase the documentation of perinatal outcomes and the aim of the article is to describe any changes in documentation before and during the implementationperiod.
My comments for the article are:
Introduction
1. Please specify in neonatal deaths and maternal mortality whether you are referring in worldwide incidence or continental incidence. It would be better to present separately the data for developing and developed countries or sub-Saharan and worldwide.
2. Provide brief explanation of the SDG3 goal.
3. Rephrase “inform interventions” on line 50.
4. Rephrase lines 60 – 63.
Materials and methods
1. The authors should explain why they chose a stepped-wedge cluster over the parallel cluster randomized trial which has lower risks of bias.
2. The authors should clarify the objective of the training in the randomized centers. Do healthcare professionals train in accurate registration of data or there is a training in acquiring skills. If the latter, briefly describe the areas of training.
3. In line 172 clarify if you included meconium-stained AF also
4. In the labor variables explain why you have not included APGAR Score in first and fifth minute of life since it is easy to calculate and has no cost. It would have been useful in the assessment of the perinatal outcomes in a future paper.
5. Please add a statement regarding the homogeneity of the sample from different regions. Do the demographic and socioeconomic factors are the same across the five regions?
Results
1. Remove lines 208 – 210.
Discussion
1. Add reference in line 355.
2. Consider adding a paragraph on the improvement of healthcare services due to the implementation of teaching strategies.
3. Add the limitations and potential biases of your research.
4. Consider adding future plans and strategies for further enhancing the documentation.
Comments on the Quality of English LanguageModerate editing of English language required
Reviewer 2 Report
Comments and Suggestions for Authors
This is a very interesting study on the topic of proper data collection and reporting in Obstetrical Units, which is unfortunately very prevalent in many if not most countries worldwide. The results of this study are interesting and applicable in many more settings even beyond Tanzania. Overall, this study is well written and I could locate no major flaws. I only have a few comments to make:
Line 25: “CQI” please use the full text terminology in the abstract.
Lines 113-140: While the authors do provide the total days of training, I miss important details on the implementation. Is the simulation training of HCW taking place throughout the given annual quarter? Are the champions and HCW trained for six and five days in total or during every quarter? Please clarify.
Lines 113-140: Additionally, since you later mention the training that the aforementioned personnel underwent with regard to the importance of recordkeeping and data acquisition (lines 329-331), you should also mention it here, since this part of the training is pertinent to the subject of this article.
Discussion: please add a strengths and limitations section. An additional brief proposal of future research prospects is also advised.
Line 355: “[ref]”: is this a missing reference? Please review and revise accordingly.
Reviewer 3 Report
Comments and Suggestions for Authors
First of all I would like to congratulate you on the work that’s been put into collecting all this data. The neo natal mortality dropping by approximately 50 % in the last 30 years is a big accomplishment worldwide. Unfortunately, as you underline in the sub-Saharan Africa the maternal mortality is still high and the sustainable development goal is not likely to be achieved.
The visits that scheduled between mentors and mentees are a great idea and the patient case reports being discussed is mandatory.
Can you please be more exact and in depth about the variables measured before and after the implementation protocol, it is obvious that documentation will be improved during and after the implementation of the monitoring in the healthcare facilities.
In my opinion the high proportion in missing data across all sites needs a national workflow to be implemented. I think that one of the eleven key obstetric indicators: gestational age is a must documentation in any admission of the patient and it’s disheartening to know that it was one of the most frequent missed documentations.
I think that your work needs to continue and hopefully through perseverance the medical system is going to have a better documentation in countries that need it.
Comments on the Quality of English LanguageThe quality of English language is good.
Round 2
Reviewer 1 Report
Comments and Suggestions for Authors
The authors addressed point-by-point the issues raised by the reviewers in the first review round. I believe that the revised manuscript describes better their work and effort.
Comments on the Quality of English LanguageMinor typing errors.